# Cloudiness Parameterization for Use in Atmospheric Models: A Review and New Perspectives

**Rae-Seol Park** [1] and **Song-You Hong** [2,3,*]

1 Korea Institute of Atmospheric Prediction Systems (KIAPS), Seoul 07071, Republic of Korea; rspark@kiaps.org
2 Cooperative Institute of Research in Environmental Science (CIRES), University of Colorado, Boulder, CO 80309, USA
3 Physical Sciences Laboratory (PSL), NOAA Earth System Research Laboratories (ESRL), Boulder, CO 80305, USA
* Correspondence: songyou.hong@noaa.gov

**Abstract:** In atmospheric models, the representation of cloudiness is a direct linkage between the moisture amount and associated radiative forcing. This paper begins by providing a review of the parameterization of cloudiness that has been used for numerical weather predictions and climate studies. The inherent uncertainties in representing a partial fraction of clouds for radiation feedback and in evaluating it against the corresponding observations are focused. It is also stated that the major hydrometeor categories of water substances such as cloud ice and water that are responsible for cloud cover are readily available in modern weather and climate models. Inconsistencies in cloud cover and hydrometeors, even in the case of the prognostic method, are discussed. The compensating effect of cloudiness for radiative feedback is found to imply that the condensed water amount itself is more influential on the radiative forcing, rather than the accuracy of the cloudiness. Based on the above perspectives, an alternative diagnostic parameterization method is proposed, utilizing a monotonic relation between the cloud water amounts and cloudiness that are obtained from aircraft and satellite observations. The basic premise of this approach lies in the accuracy of the water substance in the models, indicating that future efforts need to be given to improvements in physical processes concerning hydrometeor properties for the accurate representation of cloud radiative feedback.

**Keywords:** cloudiness; cloud fraction; cloud-radiation interaction; parameterizations

## 1. Introduction

Cloud information in atmospheric forecasting models is a key variable for computing the radiative forcing, although it is generally believed to be one of the most uncertain components. This significance of cloud information on weather and climate models is due to a tight interaction between clouds and radiation processes. The interaction also influences long-term global climate precipitation (e.g., [1,2]) and can change the local and global radiation balances. In turn, variations in radiation can affect the formation and dissipation of clouds [3]. Therefore, the accuracy and adequacy of the cloud information calculated in weather and climate models are some of the components dominating predictability. Cloud water contents and their fractional area (cloudiness) are primarily used for quantifying cloud radiative forcing; consequently, the cloud radiative feedback changes the spatiotemporal distributions of temperature. The cloud water content is computed from moist physics algorithms and is used in determining the cloud radiative properties, such as the clouds' optical thickness, the emissivity of infrared radiation by the clouds, and the effective radius of the condensed waters. For an example, the optical thickness of clouds for shortwave radiation can be expressed as

$$\tau_c = \frac{3}{2} \frac{q_c{}^{in} \Delta p / g}{\rho_c r_c}, \qquad (1)$$

where $\rho_c$ and $r_c$ are the density and effective radius of clouds in the liquid or ice phases, respectively; furthermore, $q_c{}^{in}$ represents the in-cloud water content in cloudy areas ($=q_c/C$), where $q_c$ is the grid-mean value of cloud water content from moist physics and $C$ is the cloudiness (0–1) in a model grid. In Equation (1), cloudiness, $C$, affects the radiation feedback directly.

As of late, cloud water content is well quantified with elaborate moisture physics schemes that have been developed by the numerical modeling community. For instance, in major operational centers, the complexity of the microphysics responsible for grid-resolvable precipitation physics has increased (e.g., [4–6]). Forbes implemented a prognostic microphysics scheme (MPS) with five-species water substances and demonstrated a crucial improvement in the forecast skill of the Integrated Forecasting System (IFS) of the European Center for Medium-Range Weather Forecasts (ECMWF) [5]. At the US National Centers for Environmental Prediction (NCEP), the operational global forecast system model (GFS) replaced the simple cloud physics scheme [7] with the comprehensive single-moment MPS having six-species water substances in 2019 [8]. Relatively, the quantification of cloudiness seems to be ambiguous and uncertain in the diverse cloudiness schemes. This ambiguity in quantifying the cloudiness seems to be partly associated with its definition, which is the percentage (0–100 or 0–1) of each pixel in satellite imagery or each grid box in models.

The representation of cloudiness in atmospheric models can be classified into two types of methods: diagnostic and prognostic approaches. The simplest diagnostic approach estimates the partial cloud fraction as a function of large-scale predictors such as the grid-mean relative humidity (e.g., [9,10]) or both relative humidity (RH) and cloud water (e.g., [11]). Another diagnostic approach employs statistical methods based on the probability density function (PDF) of the sub-grid-scale distributions of cloud amounts (e.g, [12,13]). The prognostic approach predicts the cloud fraction as a prognostic variable using an equation considering the sources and sinks of the cloud amount (e.g., [14]). Additionally, there are other types of cloudiness parameterizations, particularly for convective clouds [15] and clouds within a boundary layer [16]. Some formulations are based on theoretical considerations, whereas others are more empirically derived from observational data. The existence of these diverse cloudiness schemes can be attributed to the efforts to progressively include more complex cloud physical processes into the parameterization of cloudiness. Furthermore, to a certain extent, unclear definitions of sub-grid-scale cloudiness seem to contribute to the development of diverse cloudiness schemes.

The purpose of this study is to review the cloudiness parameterization schemes that have practically been used in weather and climate models. Specific attention is given to uncertainties in the parametrization and the evaluation of the predictability of the models. This paper is organized as follows: In the next section, existing cloudiness parameterization schemes are documented in brief. Section 3 presents uncertainties in cloudiness in both models and observations, along with the perspective on radiation feedback. In Section 4, an alternative approach based on the uncertainties is proposed, and Section 5 provides a summary and a conclusion of the study.

## 2. Classification of Cloudiness Parameterizations

A brief description of the existing cloudiness schemes is given here to help clarify the uncertainties in the cloudiness parameterizations in the following section. Our attention is given to the practical issues in numerical weather prediction (NWP) or climate models. A comprehensive overview of the cloudiness parameterization is available in [17].

### 2.1. Diagnostic Approaches

Diagnostic approaches can be classified according to RH schemes and a statistical scheme based on a PDF. Slingo proposed a diagnostic relationship between the RH and cloud fraction [9,18], which can be expressed by

$$C = \left( \frac{RH - RH_{crit}}{1 - RH_{crit}} \right)^2.$$
(2)

Another, similar concept was devised by Sundqvist et al. [10],

$$C = 1 - \sqrt{\frac{1 - RH}{1 - RH_{crit}}},$$
(3)

where $RH_{crit}$ defines the magnitude of the humidity variance, which is less than 1. A critical RH is set so that sub-grid-scale fluctuations allow the cloud to form when RH < 1. The cloud forms in relatively dry conditions as it becomes smaller.

Xu and Randall showed that the mean RH is a poor predictor of upper-level cloudiness in convectively active regions [11]. Upper tropospheric anvil clouds are likely to be better represented by the grid-averaged mixing ratio of condensates as the primary predictor and grid-averaged relative humidity as the secondary predictor, which is expressed as

$$C = RH^p \left[ 1 - \exp \left( \frac{-\alpha_0 q_c}{(q_s - q_v)^\gamma} \right) \right] \gamma,$$
(4)

where $p$, $\alpha_0$, and $\gamma$ are tunable constants of the scheme.

Another diagnostic approach employs the statistical schemes utilizing a specified distribution of humidity (and/or temperature) variability, which is represented by a PDF, at each grid box (Figure 1). If the PDF form for total water, $q_t$, is known, then the cloud cover is simply the integral over the part of the PDF for which $q_t$ exceeds $q_s$ (saturated water) and is given as follows:

$$C = \int_{q_s}^{\infty} G(q_t) dq_t.$$
(5)

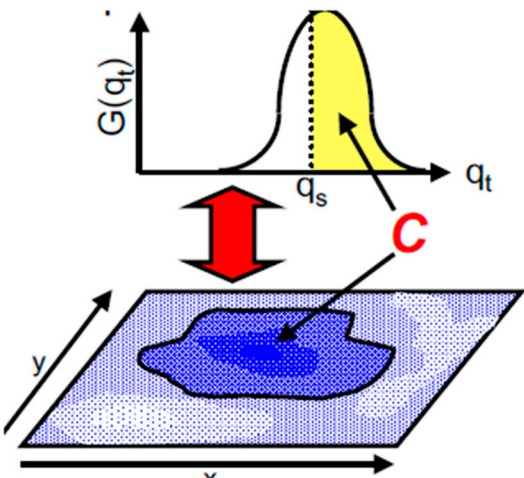

**Figure 1.** Schematic showing the statistical scheme approach. The upper panel shows an idealized PDF of the total water ($q_t$). The vertical line represents the saturation mixing ratio $q_t$ _$q_s$; thus, all the points under the PDF to the right of this line are cloudy. The integral of this area translates to the cloudy portion of the grid box, marked on the lower part of the figure, with darker shading schematically representing high total water values. Adopted from [17].

The main ingredient of the statistical schemes is therefore to give a suitable form for the PDF of total water fluctuations and derive its defining moments. Various distributions have been used: symmetric triangular PDF [12], Gaussian [19], log-normal [15], and beta [13]. The differences in the PDF shapes reflect the observational characteristics of the various cloud condensates.

Tompkins argued that there is a lack of clear distinction between the so-called "RH schemes" and statistical schemes [17]. He proved that the RH scheme of Sundqvist et al. [10] can be derived by assuming a uniform distribution for total water, and that the Smith scheme [12] also reduces to an equivalent RH formulation.

### 2.2. Prognostic Approaches

Tiedtke developed a prognostic representation of cloud cover using a budget equation [14], which can be expressed as

$$\frac{\partial C}{\partial t} = A(C) + S(C)_{CV} + S(C)_{BL} + S(C)_C - D(C) \qquad (6)$$

where $A(C)$ represents transports of cloud area; $S(C)_{CV}$, $S(C)_{BL}$, and $S(C)_C$ indicate the formations of cloud area by convection, boundary layer turbulence, and condensation processes, respectively; and $D(C)$ is the rate of decrease in cloud area due to evaporation. This equation is coupled with the convection scheme proposed by Tiedtke [20], in which the source of cloud water content $S_{CV}$ is a function of the cloud water content in updrafts and the detainment of mass. $S_{BL}$ is a function of the turbulent mass fluxes in the PBL scheme. $S(C)c$ is the formation of clouds by non-convective processes, which is parameterized as the rate of reduction in saturated specific humidity, $q_s$. The evaporation of clouds, $D$, is expressed as the rate of increase in $q_s$.

A major advantage of using prognostic equations for the cloud condensates is that cloud-driven radiation processes can be treated more physically. Based on Tiedtke's prognostic equation [14], prognostic cloudiness schemes have been suggested with an adequate modification (e.g., [21–23]). Fowler et al. implemented a more sophisticated prognostic cloud scheme into the Colorado State University GCM and improved the cloud radiation feedback [21]. Park et al. developed a prognostic cloudiness scheme [22] using Tiedtke's prognostic equation [14] and applied it to the Korean Integrated Model (KIM) [24]. They suggested a diagnostic approach using satellite observations to estimate convective cloud fraction; thus, the source term of the convective cloud fraction was replaced with a newly suggested diagnostic term in Tiedtke's prognostic equation [14]. This modification is based on the difference in CPSs in the KIM and the IFS (see the details in [22]) because a coupling with the convection parameterization is important for obtaining a realistic cloud fraction [25,26]. Muench and Lohmann also developed a new prognostic cloudiness scheme for the ECHAM-HAM (European Centre Hamburg Model–Hamburg Aerosol Model) global climate model based on Tiedtke's prognostic equation [14,23]. In their new cloudiness scheme, subsaturation and supersaturation with respect to ice are separately allowed in the cloud-free and cloudy air, and the convective condensate is always detrained.

It has been reported that the prognosed cloud fraction shows slightly better agreement with the observations than the diagnosed cloud fraction [22,27]. Recently, an issue with consistency between the cloud fraction, cloud water content, and the radiative process has been raised [28]. Since both condensed water and cloudiness are advected separately, inconsistencies between the two variables are inevitable. In KIM, the cloudiness is set to zero when condensed water does not exist. Likewise, a minimum value of cloudiness is applied to the grid points having condensed water greater than zero.

### 2.3. Practical Issues

As documented by Tompkins [17], a major drawback of the diagnostic approaches lies in the fact that the link between cloud cover and local dynamical conditions is vague. Convection certainly produces clouds if its local moistening effect is sufficient to increase the RH past the critical threshold; however, it is apparent that a grid cell with 80% RH undergoing deep convection is likely to have different cloud characteristics than a grid cell with 80% RH in a frontal stratus cloud. In atmospheric models, a distinction is made between the convective and stratiform cloud amounts, where they are represented by the

CPS and MPS, respectively. In the case of RH-based diagnostic schemes, the total cloudiness is represented as

$$C_{total} = C_{cps} + C_{mps}, \tag{7}$$

where $C_{cps}$ is the cloud fraction associated with convective clouds due to CPS, which is empirically determined as a function of the precipitation rate in CPS [9]. $C_{mps}$ is the cloud fraction as defined in Equations (2)–(4). When cloud water is prognosed by MPS, the detrained amount of convective clouds due to CPS will be the source of microphysics terms; then, the addition of $C_{cps}$ is no longer needed, as in the case of NOAA's current operational GFS model. This approach should include an assumption that convective clouds formed by CPS are fully detrained to grid points. However, if parts of the condensed clouds in CPS are detrained to the grid points and the rest remain in the sub-grid-cell, $C_{cps}$ should still be considered. This issue made the prognostic approach in the KIM, modified with a relation between the cloud fraction and cloud water amount for calculating $C_{cps}$.

In major operational centers, it is a widespread practice to tune the cloud schemes when adapted to a model. For instance, in the GFS model prior to Formula (4) of [11], the $RH_{crit}$ in Equation (2) was tuned to be a function of the land-sea mask, latitudes, altitudes, and seasons. The formula in Equation (4) is currently operational in the GFS as the cloud water and ice contents are predicted by MPS; $p$, $\alpha_0$, and $\gamma$ are required to be re-tuned when the model version is changed. In the PDF scheme, the mean and width of a given distribution form are in question. The higher moments, such as variance, can be parameterized with turbulent properties [29] by including additional tunable parameters such as the dissipation time scale of the turbulence. In the prognostic approach, the generation of sub-grid-scale clouds via cumulus convection ($S_{cv}$) or boundary layer processes ($S_{bl}$) encounters inherent uncertainties that are rooted in the parameterization of the sub-grid-scale phenomena and in their complicated interactions with each other.

## 3. Uncertainties in Cloudiness

### 3.1. Parameterizations

The diagnostic approach uses the total water substance, $q_t = q_v + q_c + q_i$ + additional condensed waters, where $q_c$ and $q_i$ are cloud water and ice, respectively. Additional condensed waters depend on the complexity of the MPSs. Fractions of clouds are determined by $q_t$ and $q_s$, where $q_s$ is the saturated specific humidity. The cloudy portion is the area greater than RHcrit in the RH schemes (see Equations (2) and (3)), where it is the integral over the saturated portion of the PDF of qt (Figure 1). A fundamental assumption in that approach lies in the fact that observed cloudiness can be represented by RH and the amount of condensed water. For convection-permitting models, this approach would be valid. As the grid size increases, however, the portion of precipitating sub-grid convection via CPS becomes larger. The diagnostic approach using RH and condensed water faces difficulty in representing the cloudiness associated with CPS. The cloud fraction from CPS highly depends upon the cloud model embedded in the CPS because the CPS is activated in convectively unstable conditions in sub-saturated columns.

The prognostic approach encounters the same issue associated with cloudiness via sub-grid-scale precipitation convection due to CPS. Park et al. formulated the source of the cloud amount ($S_{cv}$ in Equation (6)) explicitly in the CPS scheme of KIM [22], whereas it is parameterized by mass fluxes of updraft parcels in [14]. Another issue in the prognostic approach lies in the fact that condensed waters for radiative forcing are explicitly available in modern atmospheric models (e.g., [22,30–32]). In other words, the condensed waters, including cloud ice, cloud water, snow, and rain, are explicitly predicted. Note that the cloud amount was diagnosed, or its amount was predicted via simple microphysics, when the Tiedtke scheme was developed in the early 1990s. Supposing cloud hydrometeors for radiation are predicted as a 3D prognostic variable with sophisticated microphysical processes, the necessity of parameterized condensation and dissipation of clouds using the tendency of saturation vapor pressure in the Tiedke approach is in question.

It is important to note that radiation sees the total cloud amount and its fraction, whereas the total condensed water is the sum of the clouds computed from various physics modules, including CPS and MPS. Recognizing that CPS is assumed to be sub-grid-scale precipitating convection that assumes the portion of cloudy area and the remaining large-scale condensate with all or nothing in the presence of saturation, applying partial cloudiness to CPS only [15] is questionable.

### 3.2. Observation

When evaluating the modeled variables, the cloudiness in the high, middle, and low layers of the atmosphere are key variables. It would be ideal to match them as closely as possible, but their definitions differ from one to another. Recognizing that the observed quantity is based on images from aircraft and satellites, there are several factors affecting the accuracy of the observation and modeled output (see [33] for a detailed discussion). Chang and Coakley found that different methods of diagnosing the cloud cover fraction from satellite imagery yield differences of 16–30% [34]. Meanwhile, cloudiness in atmospheric models can be readily available on a specific model grid at a given level. Further, cloud overlapping needs to be applied when the satellite observation is compared with the corresponding modeled output. These uncertainties in the observed cloud fractions are likely to contribute to the difficulty in developing an accurate representation of cloudiness for radiation feedback. Concerning these uncertainties in observations, quantitative evaluation of the modeled cloudiness is less meaningful. Nevertheless, it is generally believed that RH is not an appropriate variable to be used for cloudiness (e.g., [11,17]). Shimpo et al. demonstrated that the amount of condensed water is highly correlated with the observed cloud cover, as compared with the grid-mean RH [35].

### 3.3. Radiation Feedback

In radiation code, the atmospheric response to radiation is a summed account after each component is computed separately over clear and cloudy areas. For example, downward shortwave flux ($F_{total}$) at a specific grid is expressed as

$$F_{total} = CF_{CLD} + (1 - C)F_{CLR},\tag{8}$$

where $F_{CLD}$ and $F_{CLR}$ are fluxes over cloudy and clear areas in a grid box, respectively. In Equation (1), one can tell that $q_c{}^{in}$ (=$q_c/C$) directly affects the radiative forcing due to partial clouds. For a given $q_c$, the cloud optical depth for radiation, $q_c{}^{in}$, decreases when $C$ is larger. Since the total radiative forcing is the sum of the computed cloud radiative forcings for $C$ (cloud area) and 1-$C$ (clear area) as in Equation (8), the grid-averaged cloud forcing to the atmosphere with larger C increases, although the in-cloud optical depth per area is smaller.

This compensating effect implies that the absolute magnitude of modeled cloudiness is not crucial. Although the magnitude of *C* does not linearly affect the radiative feedback, on average, a larger *C* induces stronger radiative forcing. Meanwhile, the correlation between the amount of condensed water, $q_c$, and the radiative feedback is more linearly compared with the cloudiness, which means that the magnitude of condensed water amounts is more influential on the radiative forcing than the cloudiness is.

## 4. Alternative Approach

Based on the perspectives that are discussed in the previous section, here we propose an alternative diagnostic approach. The proposed approach utilizes the statistical relationships between cloud cover and hydrometeor amount that are derived from satellite and aircraft observations. Gultepe and Isaac developed formulas showing relations between cloud fraction and cloud water content using the aircraft data from the Alliance Icing Research Study field campaign in Ontario during the winter of 1999–2000 [36]. This information is utilized to represent stratiform clouds that are predicted by MPS. The formula of the source term in representing convective clouds in [22] is utilized to estimate the cloudiness due to CPS activation. Supposing the major categories of clouds, such as cloud ice and

water, that are responsible for cloud cover are predicted, the key parameters for radiation feedback, such as cloud emissivity or effective radius, can be realistically formulated. The basic premise of this approach lies in the accuracy of the hydrometeor properties.

### 4.1. Methodology

Here we combine the cloudiness formula for stratiform and convective clouds via Gultepe and Isaac [36] and Park et al. [22], respectively, which can be expressed as

$$CF_{Lx \text{ km}} = CF_{S,Lx \text{ km}} + CF_{C,Lx \text{ km}}, \tag{9a}$$

where

$$CF_{S,Lx \text{ km}} = \frac{1}{100 - 10}[(100 - Lx)CF_{S,10 \text{ km}} + (Lx - 10)CF_{S,100 \text{ km}}] \tag{9b}$$

$$CF_{C,Lx \text{ km}} = \frac{1}{100 - 50}[(100 - Lx)CF_{C,50 \text{ km}} + (Lx - 50)CF_{C,100 \text{ km}}] \tag{9c}$$

where $CF_{S,Lx \text{ km}}$ and $CF_{C,Lx \text{ km}}$ are the stratiform cloud fraction and convective cloud fraction, respectively, in a horizontal resolution of Lx; $CF_{Lx \text{ km}}$ (the total cloud fraction) describes the sum of the cloud fractions. Here, we define the stratiform and convective clouds, which are generated by MPS and CPS, respectively. Figure 2 shows that the stratiform cloud fraction (CFs) is larger than the convective cloud fraction for a given cloud water content, implying that the convective clouds are much denser than the stratiform clouds in a specific grid. The grid-size-dependent terms in Equation (9b,c) are expressed as follows,

$$CF_{S,10 \text{ km}} = 5.57q_{c,10 \text{ km}}^{0.78}, \tag{10a}$$

$$CF_{S,100 \text{ km}} = 4.37q_{c,100 \text{ km}}^{0.77}, \tag{10b}$$

$$CF_{C,50 \text{ km}} = 5.77q_{c,50 \text{ km}}^{1.07}, \text{ and} \tag{10c}$$

$$CF_{C,100 \text{ km}} = 4.82q_{c,100 \text{ km}}^{0.94}, \tag{10d}$$

where the length scales (km) noted in the subscripts indicate the averaging scales of the observations and qc is the mixing ratio of the cloud water content (g/kg).

It is noted that the relationship shown in [36] and [22] is widely scattered. The scattered distributions imply that the relationships vary with height, which could be due to different vertical grid sizes with different heights. For this implication, the concept of cloud overlapping is borrowed. If the grid-mean cloud water content is the same in two grid points in which the grid sizes are horizontally the same but vertically different, the cloud fraction in the vertically extended grid could be greater considering the vertical overlapping. Therefore, the cloud fraction increases with height for a given grid-mean cloud water content because the vertical grid spacing in most atmospheric models expands with height. Thus, the relationship between the cloud fractions and cloud water contents (Equation (10)) can be rewritten, considering the vertical variations in cloud fraction for a given cloud water content, as:

$$CF_{Lx \text{ km}} = \alpha(CF_{S,Lx \text{ km}} + CF_{C,Lx \text{ km}}), \tag{11}$$

where $\alpha$ is a scaling factor at a given model level, which is a tunable parameter. ERA5 reanalysis was used for estimating the scaling factor. Here, the cloud water content and cloud fraction from ERA5 were applied to Equations (9)–(11), and $\alpha$ was estimated as seen in Figure 3. The computed factor increases with height because the grid depth increases upward.

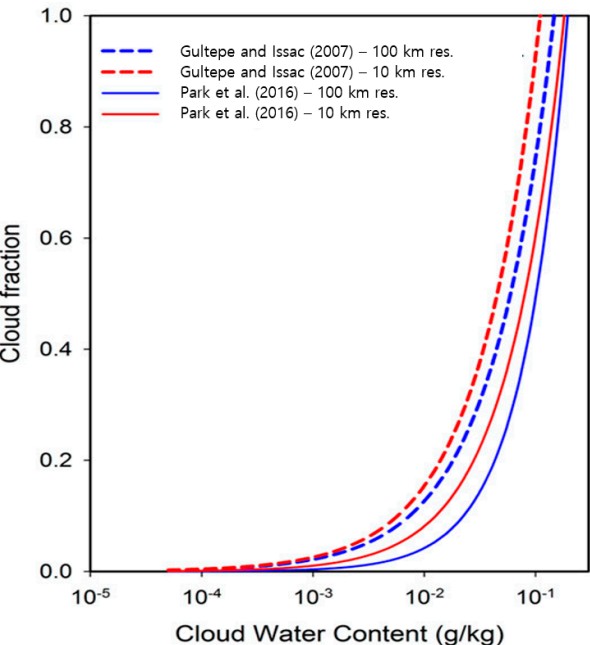

**Figure 2.** Relationships between the cloud water content and the cloud fraction derived from [22,36]: the dashed and solid lines indicate the relationships between the cloud water content and cloud fraction for stratiform clouds and convective clouds, respectively. From [22]. ©American Meteorological Society. Used with permission.

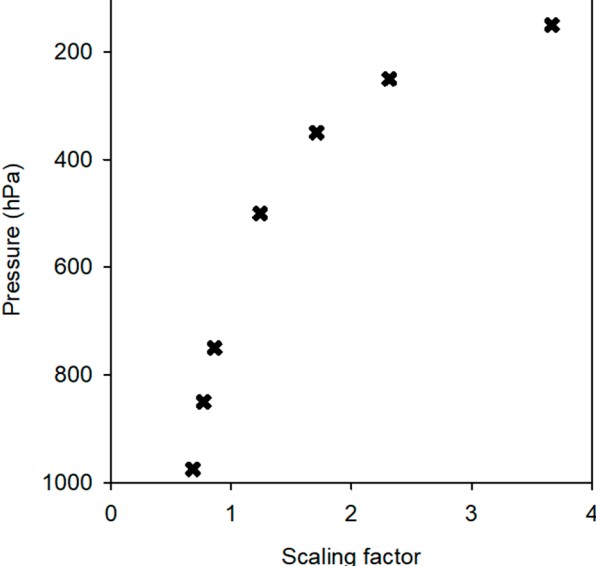

**Figure 3.** Vertical variation in scaling factors with height.

The proposed method in Equation (11) guarantees consistency between the cloud water content and cloud fraction. Moreover, one of the advantages of the alternative approach is that the change in hydrometeors due to the improvement in moist physics schemes is directly reflected into cloudiness because of the tight and explicit relationship between the cloud hydrometeor and cloud fraction.

*4.2. 1-D Tests*

The alternative approach was compared with other cloudiness schemes via 1-D tests. The 1-D tests were set to examine the fundamental characteristics of the vertical profiles of cloudiness, which are computed via the different cloudiness schemes proposed by [11,22]

and this study. For the tests, the KIM, which is developed by the Korean Institute of Atmospheric Prediction Systems (KIAPS) for the KMA's (Korea Meteorological Administration) operational use, was partly utilized to obtain data describing representative atmospheric conditions at two specific regions. One is the Equatorial Pacific region, where the convective clouds (CPS-originated clouds) are well generated with a few stratified clouds, and the other is the Northern Pacific region, where the stratified clouds (MPS-originated clouds) are dominant. The representative vertical variations in relative humidity, temperature, and cloud water content at the two specific regions were obtained via those horizontal averaging over the Northern Pacific (40 N 150 E–50 N 150 W) and Equatorial Pacific regions (10 S 150 E–10 N 150 W). Because of a lack of observed data on specific cloud hydrometeors, the representative vertical profiles were extracted from the KIM-simulated results, which are three-day forecasts because the KIM simulation begins with zero clouds. The spatially averaged vertical profiles of RH and cloud water content over the two specific regions are shown in Figure 4a,b,d,e. The prognostic cloudiness scheme described by Park et al. [22] was already adapted to KIM; thus, additional simulation is not necessary for computing the prognostic cloud fraction and the results are found in Figure 4c,f. The vertical profiles of the cloud fractions computed from [11] and this study (Equation (11)) are also shown in Figure 4c,f.

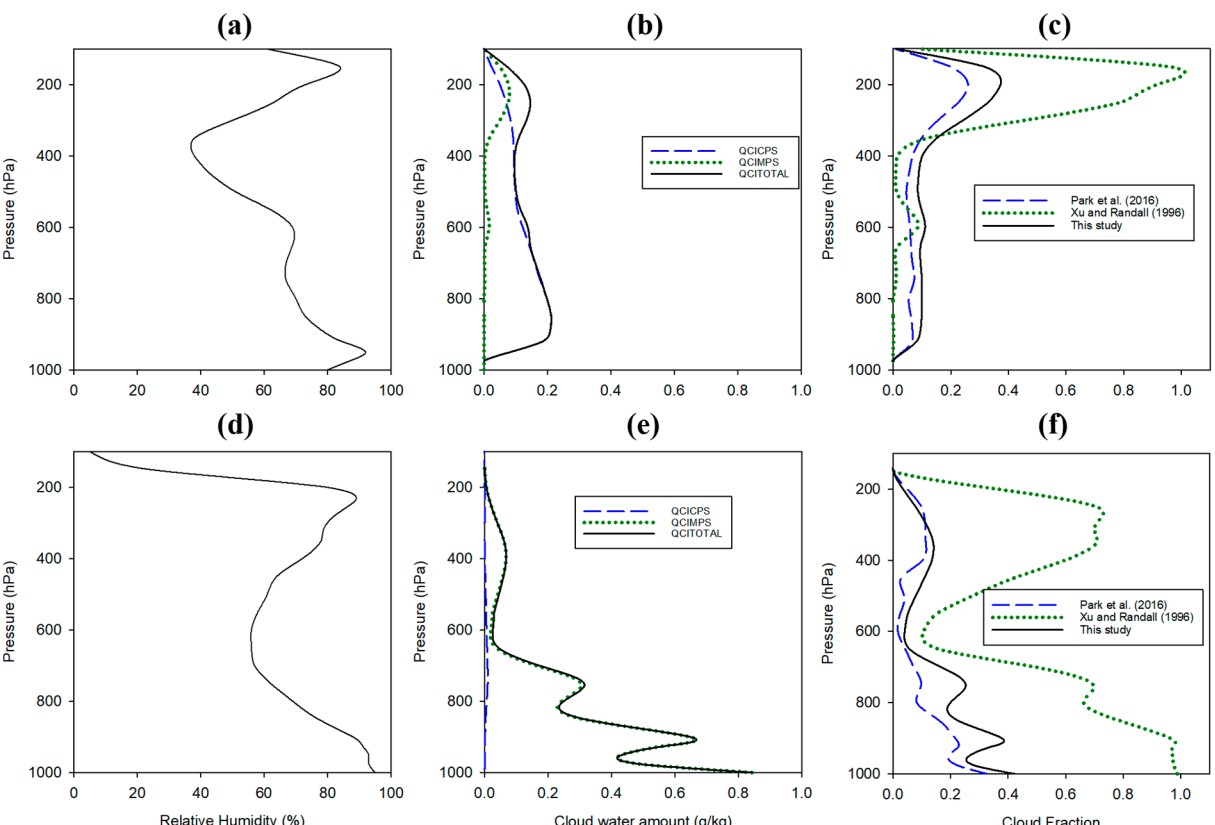

**Figure 4.** Vertical variations in (**a**,**d**) relative humidity, (**b**,**e**) cloud water amounts, and (**c**,**f**) cloud fraction; (**a**–**c**) represent the Equatorial Pacific region; (**d**–**f**) represent the Northern Pacific region.

The newly proposed approach and the prognostic cloudiness scheme by Park et al. [22] made similar patterns of cloud fractions with high correlations with total cloud water amounts; however, the pattern of cloud fraction derived from [11] is markedly different from the others (Figure 4c,f). In Figure 4c, the lower cloud fraction below 400 hPa could be attributed to missing sub-grid-scale cloud information in Xu and Randall's algorithm [11]. In Figure 4f, the relatively large cloud fraction by Xu and Randall [11] at high altitude seems to be caused by the high dependency of relative humidity. On the other hand,

the alternative approach shows a good correlation with the cloud water content and is quantitatively comparable to the prognostic cloud fraction by Park et al. [22].

## 5. Concluding Remarks

In this study, the principal uncertainties in representing cloudiness in atmospheric models and in evaluating it against the corresponding observations are discussed, followed by a concise overview of the existing cloudiness schemes. In parameterizing the cloudiness, either diagnostic or prognostic methods suffer from fundamental difficulties in the accuracy of the cloudiness. The definition of cloudiness is stated as a percentage of the cloudy area in a pixel of aircraft and satellite imagery, whereas the cloudiness in models is defined as a fraction of the cloudy area in a grid cell corresponding to the observation. In observations, the decision regarding whether it is cloudy or cloud-free is ambiguous in less dense clouds. On the other hand, the existence of clouds in models can definitely be proved with the amount of liquid and ice cloud water, but the cloud fraction is still uncertain and diverse, depending on the parameterization methods. In addition to these uncertainties, the compensating effect of cloudiness for radiative feedback implies that the condensed water amount itself is more influential on the radiative forcing rather than the magnitude of cloudiness.

Based on the uncertainties of cloudiness in both observations and parameterization in models, an alternative diagnostic approach representing cloudiness being a monotonic relation to condensed water amounts is proposed. The relation is derived from aircraft and satellite observations. Although a detailed analysis of the proposed cloudiness and quantitative impact on the model performance are not discussed in this study, the alternative approach demonstrates a reasonable pattern of cloud fraction, as compared with the prognostic approach of Park et al. [22]. Moreover, a better consistency between the cloud fraction and cloud water amount can be guaranteed because the cloud fraction in the proposed approach is exclusively based on the cloud water amount. The new method was developed using relations from limited observations and reanalysis and could be revised if more qualified observations are available in the future. The newly proposed diagnostic cloudiness scheme is currently being tested in the KIM for operational use by replacing the prognostic cloudiness scheme of Park et al. [22].

Finally, it is common practice to match the model-produced radiation fluxes at the top of the atmosphere with satellite observations. Owing to the uncertainties in cloudiness in observations and models, the quantitative evaluation of cloudiness is less reliable. As the major cloud variables for radiation, including cloud water, cloud ice, and snow, are available in most NWP and climate models, improvements in physical processes concerning cloud properties are crucial for the accurate representation of cloud radiative feedback.

**Author Contributions:** Conceptualization, methodology, software, validation, formal analysis, investigation, resources, data curation, R.-S.P. and S.-Y.H.; writing—original draft preparation, S.-Y.H.; writing—review and editing, S.-Y.H. and R.-S.P.; visualization, R.-S.P.; supervision, S.-Y.H.; project administration, R.-S.P.; funding acquisition, R.-S.P. All authors have read and agreed to the published version of the manuscript.

**Funding:** This work was carried out through the R&D project "Development of a Next-Generation Numerical Weather Prediction Model by the Korea Institute of Atmospheric Prediction Systems (KIAPS)", funded by the Korea Meteorological Administration (KMA2020-02212).

**Data Availability Statement:** The data presented in this study are available upon request.

**Acknowledgments:** The authors want to express their gratitude for valuable comments from reviewers. The second author is grateful for long-term discussion with Jimy Dudhia and Jian-Wen Bao.

**Conflicts of Interest:** The authors declare no conflict of interest.

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
