# Peer review of "Cloudiness Parameterization for Use in Atmospheric Models: A Review and New Perspectives"

_2674-0494, doi:10.3390/meteorology2030018_

Round 1
Reviewer 1 Report
Manuscript ID: meteorology-2344109, Titled: “Cloudiness Parameterization for use in Atmospheric Models: Uncertainties and an Alternative Approach” by
Rae-Seol Park and Song-You Hong
The manuscript provides a concise overview of both diagnostic and prognostic cloudiness parameterizations. It also clearly defines sources of uncertainties fir both approaches. However, when it comes to Section 4 and the “Alternative Approach” it is really not clear what exactly is the “alternative” approach. The authors reference work by Gultepe and Isac (2007) and Park et al. (2016), and in Section 4 it is unclear which part is coming from the previous studies and which part is novel. Is it only the vertical treatment new? Also, additional confusion is introduced at the begging of the Methodology section 4-a by referencing section 2-c, which does not exist.
This is important topic and an interesting study. I understand that the journal encourages shorter contribution but in order to make a stronger argument I recommend adding some more robust results (e.g. a couple of case studies). Again, it is not completely clear of what the new approach is except that closely relates to the ones represented in Park et al. 2016, therefore preliminary evaluation shows a good agreement between the two versions.
Recommendation:
I recommend the authors improve Section 4 and add more results before accepting the contribution for publication.
Reviewer 2 Report
This paper provides a useful and readable review of diagnostic and prognostic cloud schemes and proposes a diagnostic approach for cloud fraction based on statistical relationships between cloud fraction and cloud cloud amount and cloud hydrometeors.
My decision for this paper is that it requires major revision before it can be published. Main points are:
1.My primary critique is the lack of evaluation of the proposed new diagnostic-statistical cloud cover parametrization against any observational data. There is only 1 result presented in figure 4 for 2 specific regions (no global assessment) and this contains no observational data - only 3 modelled results. A similar analysis to that presented in Park et. al. (2016 -Fig 4) of cloud cover vs observations and investigation of improvements to model biases (SW, LW cloud focing and temperature fields) would seem to be a minimum to make this study publishable.
2. Motivation: The diagnostic approach advocated in this paper, a statistical relation based on observations between cloud amount and cloud hydrometeors, was actually outlined in the development of the prognostic scheme used in KIM by Park et. al. (2016). What is the motivation for now switching from the prognostic scheme to this diagnostic scheme? Cost or accuracy? Neither is clear from the text or in case of accuracy backed up by any evaluation against observations (point 1).
3. I’m unclear what the scientific advance is here - the discussion in the paper is around uncertainties in current approaches in diagnostic and prognostic cloud schemes, but the relationships used in the new diagnostic scheme in this paper are also subject to very large uncertainties as shown by the large scatter in the observational Cloudsat data (see fig 1 of Park et. al. (2016)) and comments in that paper that CloudSat cloud water contents used have biases of +/- 40% vs in situ measurement. Also, the stratiform cloud relationships are derived from one field campaign (AIRS) for one season over Ontario in winter (Gultepe and Isaac(2007)) - can this be representative of global cloud regimes? An advance on Park et. al. (2016) would have been to use additional observational data and further field campaigns available between 2016 and present to explore the efficacy of these cloud-fraction and cloud hydrometeor relationships for different global cloud regimes?
4. Vertical scaling factor - this seems very empirical and based on the modelled relationship between cloud fraction and cloud water content in the ECMWF model (ERA data). Is this scaling factor dependent on the vertical levels used in the model/re-analysis? What do the results look like without this scaling factor (at upper levels this factor is x4 which seems like a large factor?). This is not narrowing down the uncertainty in cloud parametrizations.
5. Minimising predicted temperature biases - this is the final point in the summary and it seems an odd point to raise this. I was unsure of the point being made by authors , other than all model systematic errors in T and q will obviously impact cloud predictions so should be minimised?
6. There are numerous typos in reference (e.g. Thompkins in place of Tompkins - line 234; Xu and Randall (1966) should be (1996) - line 333).
Round 2
Reviewer 1 Report
In the original review of the manuscript the major concern was lack of results. In the revised manuscript the authors indicate that the results will be available in late 2023 after the pre-implementation testing of the alternative approach within KIM is completed. I suggest that once these results are available in late 2023 (or if the authors decide to perform more robust testing in the meantime) the authors resubmit the manuscript. In other words, I don't think there is value in publishing the manuscript in its current form.
Reviewer 2 Report
The authors have addressed my points mainly by casting the paper as a review paper rather than as a detailed investigation of their new diagnostic cloud scheme. I am reasonably happy with this approach and as stated in my original review there is merit in the review aspect of cloudiness parametrization.
The authors have addressed my remaining concerns around the point 2 on the motivation to move from a fully prognostic scheme of Park et al (2016) to a new diagnostic approach, citing inconsistencies between cloud amount and hydrometeors that arise through separate advection of these quantities in the prognostic approach.
Minor revisions from my perspective would just be final check by editors on references and some of the English sentence structures.
Round 3
Reviewer 1 Report
I did miss that the article was reclassified.... Yes, I agree for the article to be published as a review.